# Development and Application of an SPR Nanobiosensor Based on AuNPs for the Detection of SARS-CoV-2 on Food Surfaces

**DOI:** 10.3390/bios12121101

**Published:** 2022-12-01

**Authors:** Leticia Tessaro, Adriano Aquino, Pedro Panzenhagen, Alan Clavelland Ochioni, Yhan S. Mutz, Paulo A. Raymundo-Pereira, Italo Rennan Sousa Vieira, Natasha Kilsy Rocha Belem, Carlos Adam Conte-Junior

**Affiliations:** 1Analytical and Molecular Laboratorial Center (CLAn), Institute of Chemistry (IQ), Federal University of Rio de Janeiro (UFRJ), University City, Rio de Janeiro 21941-909, RJ, Brazil; 2COVID-19 Research Group, Center for Food Analysis (NAL), Technological Development Support Laboratory (LADETEC), University City, Rio de Janeiro 21941-598, RJ, Brazil; 3Laboratory of Advanced Analysis in Biochemistry and Molecular Biology (LAABBM), Department of Biochemistry, Federal University of Rio de Janeiro (UFRJ), University City, Rio de Janeiro 21941-909, RJ, Brazil; 4Nanotechnology Network, Carlos Chagas Filho Research Support Foundation of the State of Rio de Janeiro (FAPERJ), Rio de Janeiro 21941-909, RJ, Brazil; 5Post-Graduation Program of Chemistry (PGQu), Institute of Chemistry (IQ), Federal University of Rio de Janeiro (UFRJ), University City, Rio de Janeiro 21941-909, RJ, Brazil; 6São Carlos Institute of Physics (IFSC), University of São Paulo (USP), São Carlos 13566-590, SP, Brazil; 7Laboratory of Immunogenetics and Molecular Biology of the General Hospital and Maternity Hospital of Cuiabá, Cuiabá 78020-840, MT, Brazil

**Keywords:** SPR, UV–Vis, COVID-19, LAMP, optical biosensor

## Abstract

A new transmission route of SARS-CoV-2 through food was recently considered by the World Health Organization (WHO), and, given the pandemic scenario, the search for fast, sensitive, and low-cost methods is necessary. Biosensors have become a viable alternative for large-scale testing because they overcome the limitations of standard techniques. Herein, we investigated the ability of gold spherical nanoparticles (AuNPs) functionalized with oligonucleotides to detect SARS-CoV-2 and demonstrated their potential to be used as plasmonic nanobiosensors. The loop-mediated isothermal amplification (LAMP) technique was used to amplify the viral genetic material from the raw virus-containing solution without any preparation. The detection of virus presence or absence was performed by ultraviolet–visible (UV–Vis) absorption spectroscopy, by monitoring the absorption band of the surface plasmonic resonance (SPR) of the AuNPs. The displacement of the peak by 525 nm from the functionalized AuNPs indicated the absence of the virus (particular region of gold). On the other hand, the region ~300 nm indicated the presence of the virus when RNA bound to the functionalized AuNPs. The nanobiosensor system was designed to detect a region of the N gene in a dynamic concentration range from 0.1 to 50 × 10^3^ ng·mL^−1^ with a limit of detection (LOD) of 1 ng·mL^−1^ (2.7 × 10^3^ copy per µL), indicating excellent sensitivity. The nanobiosensor was applied to detect the SARS-CoV-2 virus on the surfaces of vegetables and showed 100% accuracy compared to the standard quantitative reverse transcription polymerase chain reaction (RT-qPCR) technique. Therefore, the nanobiosensor is sensitive, selective, and simple, providing a viable alternative for the rapid detection of SARS-CoV-2 in ready-to-eat vegetables.

## 1. Introduction

According to the World Health Organization (WHO), more than 600 million cases of infection and 6.5 million deaths have been reported worldwide since the beginning of the coronavirus disease 2019 (COVID-19) pandemic caused by the SARS-CoV-2 virus [1]. Transmission of SARS-CoV-2 may occur from person to person, mainly by the transmission of aerosols in respiratory droplets (e.g., coughing, sneezing, talking) [2] and fomites [3]. The persistence of active viruses in fomites varies among surfaces such as plastic, steel, wood, foams, and glass [4]. For these afore mentioned surfaces, studies prove the viability of the virus, ranging from 2 h to 9 days in different conditions of temperature, ambient humidity, and specific virus type [5,6,7]. However, alternative transmission routes need to be properly investigated.

Transmission by the food route has been demonstrated, clarifying that the virus can persist in conditions found in frozen foods, packaging, and cold chain products, associated with cases of outbreaks [8]. Coronavirus has been detected in frozen chicken wings from Brazil [9]. Cases in cold chain foods such as seafood in Beijing [10], pork and beef [11], and food packaging 6 also have been found in other Chinese regions. Contamination of food samples can occur during processing, transportation, or preparation. Studies have been conducted to clarify the persistence of the virus in different types of foods that are kept under refrigeration (4 °C) and frozen (−10 °C to −80 °C). It has been shown that the SARS-CoV-2 virus can survive for up to 21 days in frozen foods and 8 days in refrigerated foods and at room temperature [12]. In foods consumed raw, such as grapes and tomatoes, the persistence of the virus was recorded for 7 days under refrigeration [2]. To our knowledge, studies involving the detection of the SARS-CoV-2 virus in refrigerated ready-to-eat vegetables have never been carried out, instead focusing only on its persistence [2].

There are four potential methods that can be employed to detect SARS-CoV-2, such as RT-LAMP, CRISPR–Cas, biosensors, and sequencing [13]. The virus concentration found in individuals and, especially, in food is relatively low, requiring prior amplification for detection. The gold-standard method used for SARS-CoV-2 virus detection is the polymerase chain reaction assay with reverse transcription (RT-PCR) [14,15]. However, the disadvantages and limitations of this technique are the prolonged analysis time and the high price of reagents and equipment, in addition to the need for trained professionals to perform the analysis. The search for the simplest, fastest, and cheapest techniques is a global demand and requires constant effort among researchers. Among the available methods, loop-mediated isothermal amplification (LAMP), developed by Notomi et al. (2000), gained prominence due to its speed, low cost, and simplicity [16,17]. The reverse transcription loop-mediated isothermal amplification (RT-LAMP) method has gained attention as it does not require state-of-the-art equipment, does not require RNA/DNA extraction or any sample preparation, and can be performed in a water bath or heating plate [18]. Visualization of amplificons can be verified by the formation of turbidity [19] with the addition of pH indicators [20,21,22,23] and fluorescent dye [24]. Combination with functionalized nanomaterials is a great advantage of the technique [25].

The main limitation of pH and fluorescent indicators is their sensitivity, since they require products generated in the reaction to visualize the result [20,21,22,23]. The use of biomolecules such as cDNA, oligonucleotide, antibodies, and aptamers improves the specificity of the biosensing method [26,27]. Using nanomaterials in biosensors improves the sensitivity, achieving satisfactory results similar to those of the standard techniques [26]. Different types of nanomaterials can be applied, but the most used are gold-based structures due to the unique properties of the metal [28]. Gold nanomaterials have an intense peak of plasmonic resonance absorption (SPR) in the visible region, which can be used to monitor the aggregation state of the nanoparticles [28]. Recently, we published a systematic review showing that the gold nanomaterial is mostly applied to colorimetric biosensors for the detection of SARS-CoV-2 [29].

In this context, we developed a biosensor based on gold nanoparticles (AuNPs) functionalized with oligonucleotides to detect SARS-CoV-2 in refrigerated, ready-to-eat vegetables (Figure 1). Samples were previously contaminated with the virus, collected from the surface with saline solution, and proceeded to the LAMP amplification stage. The functionalized AuNP solution was added to the LAMP amplicons. Positive samples had the SPR band shifted from 520 nm to ~300 nm. The nanobiosensor system has been designed to detect a region of the N gene in a dynamic concentration range of 0.1–50 × 10^3^ ng·mL^−1^. The limit of detection (LOD) was 1 ng·mL^−1^, indicating excellent sensitivity with absorbance. Our system showed 100% sensitivity when compared to the standard RT-qPCR technique.

## 2. Methodology

### 2.1. RT-LAMP

Samples of inactivated SARS-CoV-2 virus were supplied by the General Hospital of Cuiabá (Brazil). The RT-LAMP primers were designed to detect the SARS-CoV-2 N gene, according to studies previously published [23,30]. Primers were synthesized by ThermoFisher Scientific (São Paulo, Brazil) (Table 1 and Figure 2). The RT-LAMP reaction consisted of 1X WarmStart® Colorimetric LAMP 2X Master Mix (NEB, Herts, UK), 16 μM FIP and BIP, 2 μM F3 and B3, 4 μM of LF and LB, 5 μL of sample, and 9 μL ultrapure water, totaling 25 μL of reaction solution. The reaction amplicons were visualized with 1% agarose gel electrophoresis by in-gel fluorescence using the fluorescence system VÜ-F (Pop-Bio Imaging Milton Hall, Ely Road Milton Cambridge, CB24 6WZ, UK).

### 2.2. Quantitative Real-Time RT-qPCR

The RNA was extracted from inactivated SARS-CoV-2 virus using the RNeasy Mini Kit (Qiagen, Germany/ID: 74004), according to the manufacturer’s instructions. The extracted RNA was stored at −80 °C until use. We used the Fluorometer Qubit^®^ 4 Applied Biosystem (Catalog Number Q33238; Thermofisher, São Paulo, Brazil), and RNA was quantified using the RNA sensitivity assay kit (Catalog Number Q32852) to measure RNA before use. The RT-qPCR was used to confirm positive samples via the TaqMan^®^ Fast Virus Step on the StepOne™ Real-Time PCR System. The reaction involved 5 μL 4x TaqMan^®^ Fast Virus Step Master Mix, 1 μL TaqMan^®^ Gene Expression Assay (PN 4331348), 5 μL of RNA, and 9 μL of ultrapure water, totaling 20 μL of solution. The amplification was performed at 50 °C for 5 min, 95 °C for 20 s, 44 cycles for 95 °C for 3 s, and 60 °C for 30 s.

### 2.3. AuNP Functionalization with Oligonucleotides

Thiol-functionalized AuNPs (average diameter 15 nm) were acquired from Sigma-Aldrich (Code 765473). The SARS-CoV-2 cDNA probe (5′-AmMC6/-TTAGGGAGCCTTGAATACACCA-3′) was synthesized by Sigma-Aldrich (Burlington, MA, USA). The AuNP solution (5 × 10^−4^ M) was added to EDC (carbodiimide hydrochloride, 0.4 mol·L^−1^) and NHS (N-hydroxysuccinimide, 0.1 mol·L^−1^) for the activation of carboxyl-terminal groups, resulting in a final layer of succinimide ester (-COO-Suc) on the surface of gold nanoparticles. Subsequently, the solution containing the oligonucleotide probe at the concentration of 1.0 × 10^−6^ mol·L^−1^, diluted in buffer solution 1.0 × 10^−3^ mol·L^−1^ PBS/MgCl_2_, was immobilized in nanoparticles for 24 h at 4 °C under agitation.

### 2.4. Transmission Electron Microscopy (TEM) Analysis

The morphology of AuNPs was confirmed through images captured by a Transmission Electron Microscope (TEM) (Hitachi HT7800, Hitachi, Japan), operating at a voltage of 100 kV. The average particle size was estimated using ImageJ software.

### 2.5. Target Hybridization

The detection was performed via the hybridization of the AuNP solution with the amplicons generated in the LAMP reaction. For this, 5 μL of AuNP solution and 5 μL of amplicon were incubated at 95 °C for 1 h. They were later placed on ice for 5 min. Hybridization was visualized by UV–Vis in a Nanodrop One/One^c^ device (Thermofisher São Paulo, São Paulo, Brazil, Catalog Number A30224).

### 2.6. Food Sample Preparation

Ready-to-eat, refrigerated vegetables (cabbage, spinach, and lettuce) were purchased at a local grocery store in Rio de Janeiro. The vegetables were decontaminated under UV light. Leaves were sliced 3 cm × 3 cm and placed on sterile plates. A solution containing 200 µL of the inactivated virus was sprayed onto the surfaces of the vegetables to simulate droplet contamination (coughing and sneezing). Subsequently, the leaf surface was washed with 0.9% saline solution to recover the inactivated virus droplets, homogenized, and stored in an ice bath for the LAMP and RT-qPCR reaction.

Figure 3 shows a summary of the main design steps of the nanobiosensor based on AuNPs for the detection of SARS-CoV-2 on food surfaces.

## 3. Results and Discussion

### 3.1. RT-LAMP Parameter Optimization

To first assess the optimal condition for the DNA amplification via the LAMP technique, a central compound design (RCCD) was employed. As the LAMP technique employs a biological enzyme, whose optimum working parameters can vary, this step was considered crucial prior to conducting the biosensor assays. Therefore, the temperature and time ranges were set as 54–76 °C and 15–65 min, based on the enzyme manufacturer’s information, and we consulted the available literature for the LAMP parameters. The adjusted RCCD model presented an R^2^ of 0.81 and a significant effect of the linear terms from time and temperature influencing the technique’s DNA yield. A response surface drawn from the adjusted model was then used to evidence the ideal conditions for the performance of LAMP (Figure 4).

Data show that the highest DNA yield was achieved at temperatures between 60 and 65 °C and reaction times greater than 35 min. As expected, the results showed the reaction to be more dependent on temperature than time, corroborating with the working temperatures of the polymerase BST 2.0 and primer annealing. Therefore, from the previous surface response, we obtained the optimum conditions for the LAMP reaction at 65 °C and 40 min based on the results.

Furthermore, the stability and useful life of the SPR biosensor is highly influenced by variations in temperature, pH, and ionic concentrations, which influence the characteristics of the bioreceptor [31]. Therefore, all reagents and enzymes were stored at −20 °C, as prescribed by the manufacturer, with a shelf life of 24 months.

### 3.2. AuNP Functionalization and AuNP-Oligo-Target Hybridization

The initial characterization of AuNPs was performed using TEM, to attest to the size and morphology of the particles. The morphology influences the absorbance of the sensor, due to the change in the position and intensity of the Pico SPR to detect binding events on metallic surfaces. Furthermore, as attested by our work [28], small-sized spherical nanoparticles are preferred when designing optical biosensors. When a nanoparticle presents sharp corners, it generates more peaks when compared to spherical ones, shifting the plasmonic resonance bands to the infrared region. Regarding the size of the nanoparticles, the more the NP diameter increases, the greater the light scattering becomes, shifting the SPR band towards the red region [32]. The choice of 15 nm AuNPs guaranteed more symmetrical and less widened bands for better monitoring in the visible spectrum. TEM confirmed the spherical shape of the particles and the mean size of 15.2 ± 1.5 nm (Figure 5A). In addition, the uniform distribution of the particle population was verified by means of a histogram of the sizes (Figure 5B).

The confirmation of the target detection, i.e., LAMP-generated amplicons, with the AuNPs, could be observed in terms of the shifting of the gold nanoparticles’ SPR band from 525 nm to ≈300 nm (see Figure 5C,D). The displacement of the SPR absorption band of the AuNPs was due to hybridization with the target amplicon, generating a change in the aggregation of the nanoparticles [33], which, in our case, indicated the detection of SARS-CoV-2. The size and aggregation of AuNPs directly affect the wavelengths in which they absorb and disperse the dielectric properties of light (refraction index) of the medium and interparticle [32].

Hybridization is a crucial step in biosensor applications, where the AuNPs functionalized with the complementary oligonucleotide can bind to the SARS-CoV-2 amplicons generated in the LAMP reaction. Since the generated amplicons are double-stranded DNA formed by the LAMP reaction, no natural hybridization would occur with the complementary oligonucleotide of the AuNP. Therefore, a heat step is necessary for the double tape to be opened (i.e., melting) and hybridization to take place, as shown in the schematic representation in Figure 3C. As the melting temperature of the amplicon varies with the product size, the adequate selection of the heating time is vital in LAMP-based sensors, as the failure of amplicon melting can lead to false negative results. Therefore, we evaluated the melting times of 30, 45, and 60 min at 95 °C, and found that at least 60 min is required for proper melting in the proposed sensor. Lastly, with all the functional parameters of the nanobiosensor optimized and checked, the sensor was then tested for its detection limit.

### 3.3. Determining Biosensor LOD for Application in Food Samples

The method limit of detection (LOD) was determined using samples of pure SARS-CoV-2 RNA. Additionally, samples containing inactivated betacoronavirus SA44, a virus from the same family as SARS-CoV-2, were evaluated, and no amplification was detected from the LAMP reaction, thus demonstrating the high specificity of the method due to the primers used in the RT-LAMP reaction, designed to amplify a specific region of the N gene 

The LOD was visually estimated from UV–Vis spectra, with a dynamic concentration range from 0.1 to 5000 ng mL^−1^ of SARS-CoV-2 viral RNA (Figure 6A). From the visual inspection, the LOD of 1 ng mL^−1^ (2.7 × 10^3^ copy per µL) was established as the lower concentration in which SPR band displacement was found. This result was further confirmed by gel electrophoresis (Figure 6B). Despite presenting amplification at the concentration of 0.1 ng mL^−1^, there was no displacement of the SPR band, similar to the negative control, indicating that the LAMP amplification product at this concentration was not sufficient to be detected by the nanobiosensor (Figure 6A). This can be understood as an indication of the insufficient amplicon concentration to change the aggregation state of AuNPs, causing the SPR band to shift.

From the promising results found for the method LOD, we further decided to evaluate our biosensor in vegetables, since they are ready-to-eat products and can help in the spread of the virus. For this, the samples were analyzed using the developed nanobiosensor and compared with the gold-standard technique, RT-qPCR. Figure 7 shows the results of the use of the nanobiosensor developed to detect SARS-CoV-2 in food.

A set of eight samples, comprising three positives and one negative sample for each high or low concentration, were analyzed. In both concentrations, the nanobiosensor was able to detect the virus in the contaminated food (Figure 7A,B), showing that the LAMP technique is sensitive and of direct use, without the need for sample extraction, elution, and concentration steps. Further confirmation was achieved by electrophoresis, which showed amplification at the two tested viral concentrations, and the generated fragment matched the target of the functionalized AuNPs (Figure 7C).

Additionally, the same sample used in the biosensor was subjected to the RT-qPCR technique, which was not able to discriminate the positive and negative samples (Figure 7D). This can be attributed to the RNA extraction not being 100% efficient. As the RT-qPCR requires specific sample extraction and preparation protocols, there is the need for a higher concentration of RNA to perform the analyses, which is one of the limitations of the technique. In comparison, the RT-LAMP technique was able to detect 1 ng mL^−1^ without the need for an RNA extraction step. The biosensor was able to detect, with 100% sensitivity, the samples of lettuce, cabbage, and spinach contaminated with the SARS-CoV-2 virus (Figure 8), at both high and low infection levels, proving to be an excellent alternative for food quality control.

In light of the novelty of the present nanobiosensor application, a comparison of studies was performed in terms of the LOD, specificity, and sensitivity of several biosensors aiming at the detection of SARS-CoV-2 (Table 2). Within all the available studies in the literature, only one study was found to apply a biosensor to detect the virus in food [34]. Zhou et al. (2022) applied a sensor to sea food in China’s cold chain, which is reported as a major source of contamination in the country. The use of a quantum nanomaterial dot@Zn-metal-organic framework (g-CNQDs@Zn-MOF) and fluorescence spectrophotometer reading ensured high sensitivity. This study promotes the development of new systems to monitor SARS-CoV-2 in cold-chain foods and ready-to-eat foods. However, we emphasize that new, simple, and low-cost alternatives should be developed, such as the one proposed in our work. Despite the lower sensitivity of our sensor, it was able to detect contamination in food surfaces at high and low concentrations, using a nanomaterial and simple equipment such as a water bath and UV–Vis reader. These advantages make it robust for monitoring food in the field, especially in resource-limited locations.

Moreover, the comparison with the literature shows that, besides being one of the only biosensors applied for the detection of SARS-CoV-2 in food, our sensor also presents an LOD, high specificity, and 100% sensitivity, in a similar range to the others.

Although our system requires a longer time when compared to the others, the LAMP amplification step guarantees high specificity, reaching 100%. These analytical parameters are extremely important, as they guarantee the reliability of the biosensor; however, they are not presented in the other studies.

The majority of applications of biosensors to clinical samples can be seen in Table 2. Taking into account the statement recently published by the WHO stating that food is one of the contamination routes of SARS-CoV-2, our study, in addition to offering an easy-to-apply methodology for monitoring SARS-CoV-2, also encourages further studies on alternative methods for the monitoring of SARS-CoV-2 in food.

## 4. Conclusions

Here, we report the development of a nanobiosensor with SPR transduction for the detection of SARS-CoV-2 on food surfaces. The biosensor is based on AuNPs, with a 15 nm diameter, functionalized with an oligonucleotide complementary to the amplicons generated by the LAMP technique. The LAMP technique was optimized using DCCR to establish the best conditions for the parameters, namely time and temperature. Assays showed that the temperature of 65 °C, applied for 40 min, generated the highest concentrations in DNA formation (amplicons in LAMP). The biosensor showed an excellent LOD of 1 ng·mL^−1^ (2.7 × 10^3^ copy per µL) and high specificity compared with another betacoronavirus. The nanobiosensor was applied to detect SARS-CoV-2 in samples of lettuce, cabbage, and spinach, with accuracy of 100%, indicating excellent discrimination in terms of the presence and absence of the SARS-CoV-2 virus on food surfaces. The standard RT-qPCR technique could not discriminate the positive samples, requiring high viral concentrations. The nanobiosensor proved to be a simple, fast (100 min), and low-cost alternative capable of ensuring food safety and virus dissemination, since food is a possible route of contamination of COVID-19.

## Figures and Tables

**Figure 1 biosensors-12-01101-f001:**
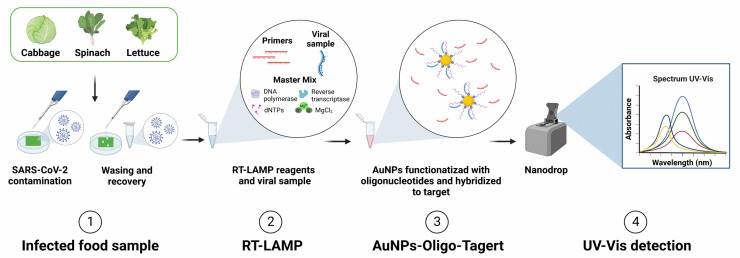
Schematic representation of the main stages of development and application of a nanobiosensor based on AuNPs functionalized with oligonucleotides for the detection of SARS-CoV-2 in food surfaces.

**Figure 2 biosensors-12-01101-f002:**
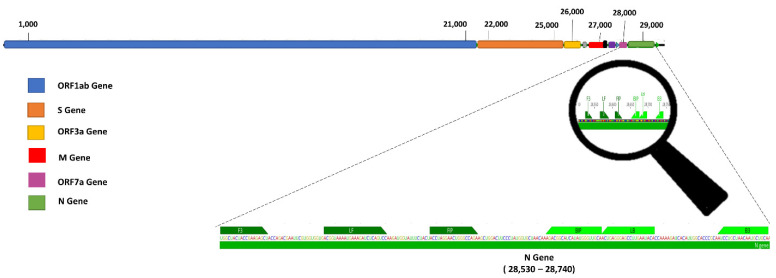
Design of N-gene-specific primer sets for SARS-CoV-2 (Table 1), designed to selectively amplify the 28,530–28,740 nucleotide segment. Their corresponding positions are demonstrated in the magnification of the image.

**Figure 3 biosensors-12-01101-f003:**
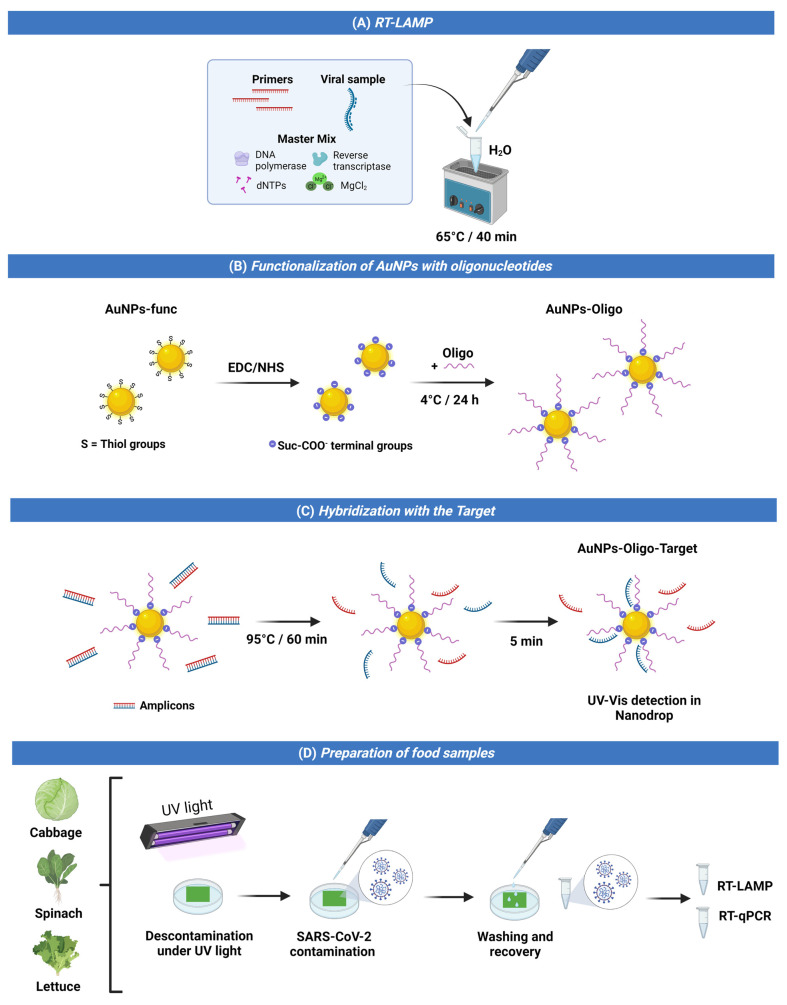
Key design steps of AuNP-based nanobiosensor for detection of SARS-CoV-2 on food surfaces: (**A**) RT-LAMP reaction and virus sample addition; (**B**) AuNPs containing thiol groups were functionalized with oligonucleotides; (**C**) separate amplicons bind oligonucleotide groups to the target for further detection in UV–Vis; (**D**) in natura samples of vegetables (cabbage, spinach, and lettuce) decontaminated with UV light were infected with SARS-CoV-2 and recovered for RT-LAMP and RT-qPCR reactions.

**Figure 4 biosensors-12-01101-f004:**
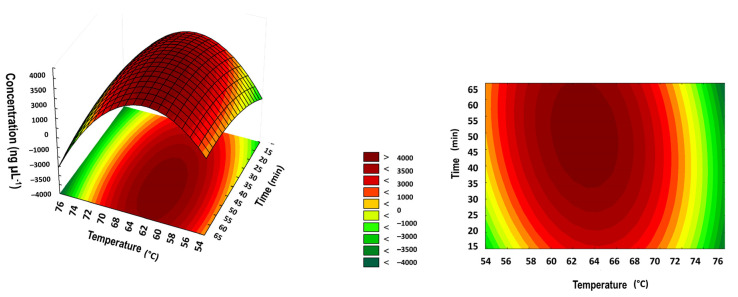
Response surface obtained from 11 experiments carried out according to the rotational central compound design (RCCD).

**Figure 5 biosensors-12-01101-f005:**
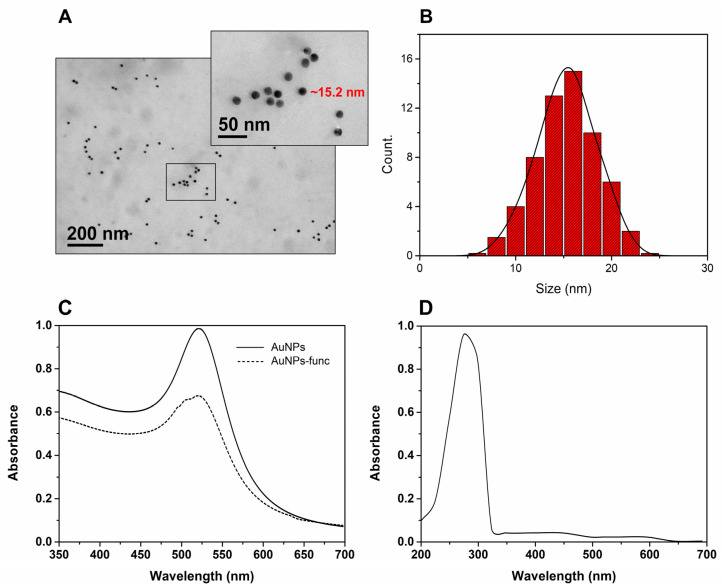
(**A**) TEM image of AuNPs indicating distribution with scale bar magnification: 200 nm and 50 nm. (**B**) Particle size distribution histogram determined from the TEM image. (**C**) UV–Vis spectrum indicates the absorbance changes between the AuNPs (solid line) and the functionalized AuNP-Oligos (dashed line). (**D**) UV–Vis spectrum indicates the displacement of the SPR band after hybridization of the target amplicon with AuNP-Oligos, evidencing the detection of SARS-CoV-2.

**Figure 6 biosensors-12-01101-f006:**
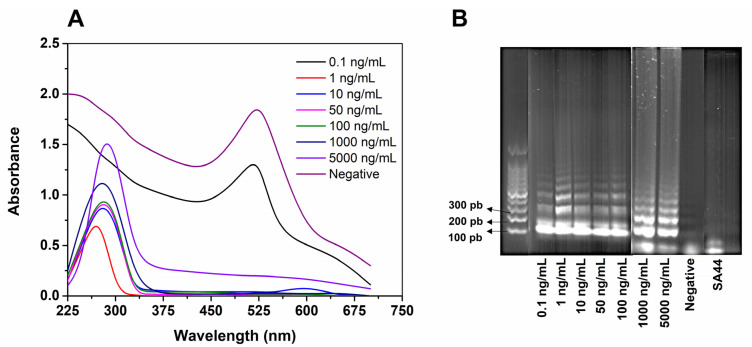
(**A**) UV–Vis spectra for dynamic concentrations ranging from 0.1 to 50 × 10^3^ ng mL^−1^ of RNA of SARS-CoV-2 virus, negative control, and betacoronavirus SA44. (**B**) Electrophoresis with 1% agarose gel confirming distinct concentration of RNA amplification and identification of amplified fragment in the target range of 100 to 200 bp.

**Figure 7 biosensors-12-01101-f007:**
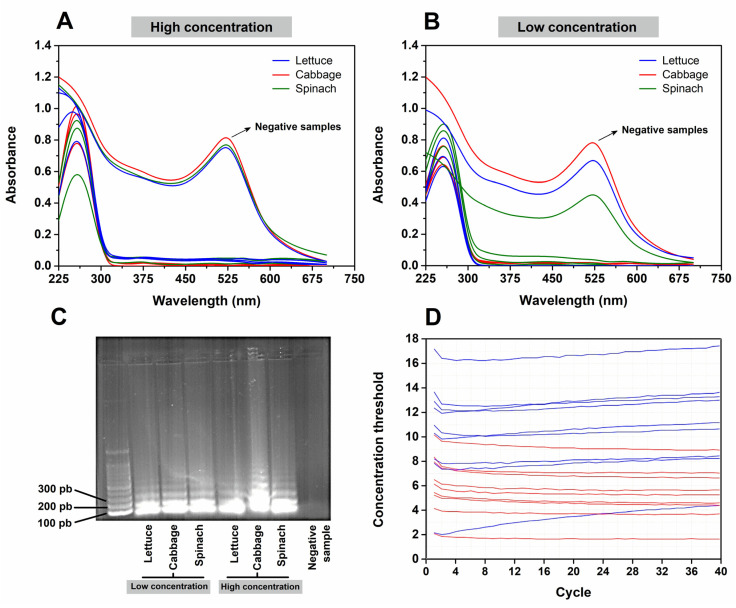
(**A**,**B**) The UV–Vis spectra after the application of the biosensor to detect SARS-CoV-2 at high (Ct:19) and low viral concentrations (Ct:31) in vegetables (lettuce, cabbage, and spinach), respectively. (**C**) Electrophoresis with 1% agarose gel to confirm RNA amplification in vegetable samples. (**D**) Image generated by RT-qPCR, showing that there was no detection in the samples subjected to the RNA extraction.

**Figure 8 biosensors-12-01101-f008:**
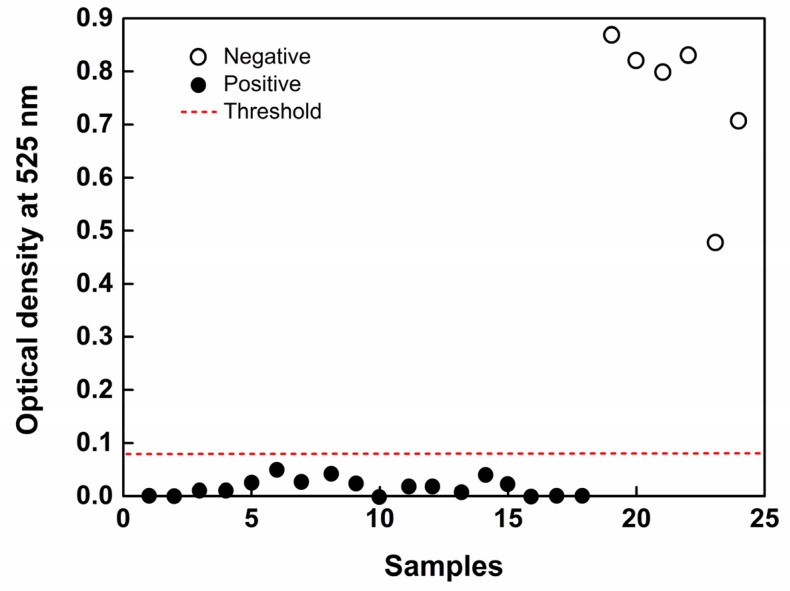
Absorbance values for food samples infected (positive) or not infected (negative) were recorded using a plate reader. A threshold line (red line) was set that maximized the assay’s discrimination performance.

**Table 1 biosensors-12-01101-t001:** Sequence of primers for RT-LAMP with target gene N according to references [23,30].

	Sequence	Pb
FIP	TCTGGCCCAGTTCCTAGGTAGTCCAGACGAATTCGTGGTGG	41
BIP	AGACGGCATCATATGGGTTGCACGGGTGCCAATGTGATCT	40
F3	TGGCTACTACCGAAGAGCT	19
B3	TGCAGCATTGTTAGCAGGAT	20
LF	GGACTGAGATCTTTCATTTTACCGT	25
LB	ACTGAGGGAGCCTTGAATACA	21

**Table 2 biosensors-12-01101-t002:** Developed biosensors for detection of SARS-CoV-2 in several real samples.

Biosensor Technology	Recognition Element	LOD	SEN	SPE	Concentration Range	Time	Sample	Reference
SPR (UV–Vis)	Oligo	1.0 ng·mL^−1^	100%	high	0.1–50 × 10^3^ ng·mL^−1^	100 min	Cabbage, lettuce and spinach	This work
Fluorescence	Aptamer	1.0 pg·mL^−1^	100%	good	5.0–1.0 × 10^3^ pg·mL^−1^	30 min	Cherry, frozen shrimp, salmon, and frozen fish	[34]
LSPR (UV–Vis)	NeuNAc	40 µg.mL^−1^	-	-	9400–6 × 10^5^ ng·mL^−1^	-	Nasal swabs	[35]
Colorimetric or UV–Vis	Antibody	48 pg·mL^−1^	-	high	250–1000 ng·mL^−1^	10 min	Saliva	[36]
SPR–colorimetric	ASOS	180 ng·mL^−1^	-	-	200–3 × 10^3^ ng·mL^−1^	10 min	Patient samples	[37]
LSPR (UV–Vis)	Antigen	150 ng·mL^−1^		*	150–650 ng·mL^−1^	10 min	Clinical	[38]
Colorimetric	Antibory	1.0 ng·mL^−1^	-	high	0.1–100 ng·mL^−1^	30 min	Throat and nose swabs	[39]
Colorimetric	ACE2	154 ng·mL^−1^	96	98	10^−3^–10^3^ ng·mL^−1^	3 min	Nasal	[40]
FET	Antibody	160 PFU mL^−1^	-	-	-	>1 min	Clinical	[41]
Electrochemical	Antibody	0.09 pM		high	1 fM–1 µM	30 s	-	[42]
PPT+LSPR	cDNA	0.22 pM	-	-	1 pM–1 nM	-	-	[43]

Legend: (*) have cross-reactivity for viruses SARS-CoV and MERS-CoV; specificity (SPE), sensitivity (SEN), field-effect transistors (FET); plasmonic photothermal (PPT); N-acetyl neuraminic acid (NeuNAc); localized surface plasmon resonance (LSPR); thiol-modified antisense oligonucleotides specific for N-gene (ASOS).

## Data Availability

Not applicable.

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
