# Peer review of "Development and Application of an SPR Nanobiosensor Based on AuNPs for the Detection of SARS-CoV-2 on Food Surfaces"

_biosensors, 2022, doi:10.3390/bios12121101_

Round 1

Reviewer 1 Report

Comments to Authors

The article entitled “Development and application of an SPR nanobiosensor based on AuNPs for the detection of SARS-CoV-2 in food surfaces” is very insightful and interesting. I would recommend for publication addressing the following comments.

1.     Number of other methods applied for the SARS-CoV-2 detection can be mentioned in introduction.

2.     The authors must brief the assay’s selectivity that works on SARS-COVID-2.

3.     The authors should also mention the sensitivity value of the assay.

4.     Compare the SARS-COVID-2 detection parameters with more references.

5.     Table of SARS-COVID-2 food analysis can be compared with other reported works.

Author Response

Dear editor Despina Kalogianni,

Please find enclosed the revised manuscript entitled Development and application of an SPR nanobiosensor based on AuNPs for the detection of SARS-CoV-2 in food surfaces”.

We thank you for managing our article. The manuscript has been severely reviewed in light of the referee's comments. We also thank the reviewers for their extremely useful assessment of this article. Our detailed responses are provided below and demonstrate that all comments significantly improved this work. Responses to the referee’s comments and changes are highlighted in red for clarity. We hope these changes prove satisfactory, although any additional suggestions are welcome.

# Reviewer 1

The article entitled “Development and application of an SPR nanobiosensor based on AuNPs for the detection of SARS-CoV-2 in food surfaces” is very insightful and interesting. I would recommend for publication addressing the following comments.

Response: Thank you for the constructive comments. We have addressed all criticisms, clarifying the issues raised point by point and implementing all suggestions to the original text.

  1. Number of other methods applied for the SARS-CoV-2 detection can be mentioned in introduction.

Response: Thank you. There are four potential methods that can be employed to detect SARS-CoV-2, such as RT-LAMP, CRISPR–Cas, Biosensors, and Sequencing. https://doi.org/10.3390/chemosensors10060221. (Line, 70-71)

  1. The authors must brief the assay’s selectivity that works on SARS-COVID-2.

Response: Thank you. The selectivity of the assay was evaluated using betacoronavirus of the SA44 strain. The assay was selective even with viruses from the same family (corodaviridae) of SARS-CoV-2. This information is reported on line 268.

  1. The authors should also mention the sensitivity value of the assay.

Response: The sensitivity of the assay is reported in line 275 (LOD 1 ng mL-1) and also a dynamic concentration range from 0.1 to 5000 ng mL-1.

  1. Compare the SARS-COVID-2 detection parameters with more references.

Response: Thank you for your observation. More references were debuted in Table 2 and discussed.

  1. Table of SARS-COVID-2 food analysis can be compared with other reported works.

Response: Thank you for your observation. More references were debuted in Table 2 and discussed.

Reviewer 2 Report

SPR nano biosensor was developed for the detection of SARS-CoV-2 using AuNPs functionalized with oligonucleotides in food surfaces. The manuscript is well organized but exists some shortcomings. I could not advice to publish the study on the journal with the current version. The manuscript needs a major revision. Here are some questions and advice in the following that may be useful for the manuscript.

1.      The text in figure is not readable. This should be change with best image. Prefer to TIFF file.

2.      Please mention the company name, address etc. of each equipment like you mentioned RNeasy 133 Mini Kit information.

3.      I will recommend the author to discuss in detail about the stability and shelf life of the mentioned SPR biosensor.

4.      Why BST.2.0 is selected? Why not 3.0 and others? Justify it

5.      Hoe about the density of the AuNPs? How much dense they are? As the numbers of nanoparticles affect on the binding and detection.

6.      I appreciate if you convert the LOD in terms of copies/reaction.

7.      Explain the LOD in terms of better way, e.g., provide statistical analysis

8.      Explain the LOD in terms of FDA guidelines. Is this biosensor being good for commercialization and scalability?

9.      Figure 7 (c) scale bar is not readable

10.  All figures should be readable and clear for the readers.

11.  How about the detection time? Also explain this parameter of a sensor.

Author Response

Dear editor Despina Kalogianni,

Please find enclosed the revised manuscript entitled Development and application of an SPR nanobiosensor based on AuNPs for the detection of SARS-CoV-2 in food surfaces”.

We thank you for managing our article. The manuscript has been severely reviewed in light of the referee's comments. We also thank the reviewers for their extremely useful assessment of this article. Our detailed responses are provided below and demonstrate that all comments significantly improved this work. Responses to the referee’s comments and changes are highlighted in red for clarity. We hope these changes prove satisfactory, although any additional suggestions are welcome.

# Reviewer 2

SPR nano biosensor was developed for the detection of SARS-CoV-2 using AuNPs functionalized with oligonucleotides in food surfaces. The manuscript is well organized but exists some shortcomings. I could not advice to publish the study on the journal with the current version. The manuscript needs a major revision. Here are some questions and advice in the following that may be useful for the manuscript.

Response: We would like to thank you for doing constructive remarks on the manuscript. All of the comments and suggestions together with the observations from the other reviewers helped to improve the scientific level of our article.

  1. The text in figure is not readable. This should be change with best image. Prefer to TIFF file.

Response: Thank you. The text of the figures has been enlarged to improve its visualization. In addition, all figures were saved in TIFF format, with a resolution of 600 dpi.

  1. Please mention the company name, address etc. of each equipment like you mentioned RNeasy 133 Mini Kit information.

Response: Thank you for the suggestion. We mention in detail all equipment and reagents, and we highlight in red the changes.

  1. I will recommend the author to discuss in detail about the stability and shelf life of the mentioned SPR biosensor.

Response: Thank you for the comment. The shelf life of the SPR biosensor was discussed in lines 218- 221.

  1. Why BST.2.0 is selected? Why not 3.0 and others? Justify it

Response: Thank you. We reproduced the exact protocol by Zhang et al., 2020 and Aoki et al., 2021. We were afraid that changes in these protocols could interfere with the normal amplification yield. So, we followed the sequence of the primers, and the BST polymerase and master mix were purchased from the same manufacturer.

  1. Hoe about the density of the AuNPs? How much dense they are? As the numbers of nanoparticles affect on the binding and detection.

Response: The density of the solution found in the vial was 1 g/mL. The number of nanoparticles present in the solution influences the intensity of the color generated after aggregation and consequently detection.

  1. I appreciate if you convert the LOD in terms of copies/reaction.

Response: As suggested, the LOD was converted in terms of copies, lines 37, 275 and 366.

  1. Explain the LOD in terms of better way, e.g., provide statistical analysis

Response: It is therefore evident that statistical methods for assessing and comparing limits of detection are of importance. In general terms, the limit of detection of an analyte may be described as that concentration which gives an instrument signal significantly different from the ‘blank’ or ‘background’ signal. This description gives the analyst a good deal of freedom to decide the exact definition of the limit of detection, based on a suitable interpretation of the phrase ‘significantly different’. There is still no full agreement between researchers, publishers, and professional and statutory bodies on this point. Taking into in mind, we estimated the LOD using Figure 6 in which the concentration of 1.0 ng.mL–1 was significantly different from the ‘blank’. We added some pieces of these sentences in the manuscript, Page 7.

  1. Explain the LOD in terms of FDA guidelines. Is this biosensor being good for commercialization and scalability?

Response: We understand that the FDA guidelines determine the LOD estimative assuming values set to 3-time standard deviations above the blank response by linear instrument response. However, as said above, we estimated the LOD according to the spectrums of Figure 6, with a concentration of 1.0 ng. mL–1 that was significantly different from the ‘blank’. Currently, this short communication focused on announcing the applicability of food samples, and in the next study, we will verify its suitability for commercialization and scalability.

  1. Figure 7 (c) scale bar is not readable

Response: Figure 7 (C) has been reformulated with an increase in the scale bar to improve visualization.

  1. All figures should be readable and clear for the readers.

Response: Thank you for the observation. All figures were saved in TIFF format, with a resolution of 600 dpi.

  1. How about the detection time? Also explain this parameter of a sensor.

Response: The system time for detection is 100 min (line 372). The detection time is a very important parameter, the goal is to design sensors that require less time than standard techniques. The standard PCR and ELISA techniques are considered time-consuming because they can reach from 3h to 9 days, and sensors that overcome this limitation are appreciable.

Reviewer 3 Report

The manuscript presents the development of a nanobiosensor based on gold spherical nanoparticles with surface plasmonic resonance transduction for the detection of SARS-CoV-2 in food surfaces. This study can be accepted for publication in Biosensors pending some revisions as follows:

(1) Developed biosensors for detection of SARS-CoV-2 in several real samples are compared in Table 2. However this table compares the detection limits and other parameters such as sensitivity and quality factor should also be considered and addressed.

(2) A scale bar is needed for Figure 5C.

(3) Is there any optimization method to design the nanobiosensor? Please clarify.

(4) The quality of some figures such as Figures 6a and 7c is poor and they should be presented in more resolution. Also, the unit of vertical axis is not clear. Is the Absorbance normalized?

(5) The results need to be verified. 

(6) Figure 4 should be presented in the text.

Author Response

Dear editor Despina Kalogianni,

Please find enclosed the revised manuscript entitled Development and application of an SPR nanobiosensor based on AuNPs for the detection of SARS-CoV-2 in food surfaces”.

We thank you for managing our article. The manuscript has been severely reviewed in light of the referee's comments. We also thank the reviewers for their extremely useful assessment of this article. Our detailed responses are provided below and demonstrate that all comments significantly improved this work. Responses to the referee’s comments and changes are highlighted in red for clarity. We hope these changes prove satisfactory, although any additional suggestions are welcome.

# Reviewer 3

The manuscript presents the development of a nanobiosensor based on gold spherical nanoparticles with surface plasmonic resonance transduction for the detection of SARS-CoV-2 in food surfaces. This study can be accepted for publication in Biosensors pending some revisions as follows:

Response: We would like to thank you for doing constructive remarks on the manuscript. All of the comments and suggestions together with the observations from the other reviewers helped to improve the scientific level of our article.

(1) Developed biosensors for detection of SARS-CoV-2 in several real samples are compared in Table 2. However this table compares the detection limits and other parameters such as sensitivity and quality factor should also be considered and addressed.
Response: Thank you for the observation. We've added other parameters for comparison that can be checked in table 2.

(2) A scale bar is needed for Figure 5C.

Response: Thank you. Scale bars were added in Figure 5(C), considering the increases at 200 nm and 50 nm.

(3) Is there any optimization method to design the nanobiosensor? Please clarify.
Response: Thank you for the question. The sensor parameters, which involved the nanoparticles followed the condition set by the work of Soares et al (2021) https://doi.org/10.1039/D1QM00665G). Furthermore, in the design of the nanobiosensor we have optimized the LAMP technique reaction parameters (temperature and time), as the LAMP technique was our implementation for the sensor (against the Genosensor of the previously cited study). The LAMP optimization is discussed in section 3.1.

(4) The quality of some figures such as Figures 6a and 7c is poor and they should be presented in more resolution. Also, the unit of vertical axis is not clear. Is the Absorbance normalized?

Response: Thank you. All figures in the manuscript have been reworked for better resolution (TIFF format at 600 dpi). Figures 6(A) and 7(C) show the readable axes with normalized absorbance values.

(5) The results need to be verified. 

Response: Thank you for the comment. The results were verified and the modifications made can be observed marked in red.

(6) Figure 4 should be presented in the text.

Response: Thank you for the observation. The figure 4 was presented in the text, in line 201.

Round 2

Reviewer 2 Report

The manuscript has been carefully revised and met the requirement of publication in the journal. I suggest accepting the manuscript in the present form.